# First trimester "clean catch" urine and vaginal swab sample distinct microbiological niches

Juliana Sung,[1] Peter Larsen,[2] Thomas M. Halverson,[3] Thaddeus P. Waters,[1] Jean R. Goodman,[1] Alan J. Wolfe[2,3]

**ABSTRACT** Untreated asymptomatic bacteriuria (ASB) has been associated with adverse pregnancy outcomes. Thus, routine screening by standard urine culture (SUC) and treatment of ASB are currently recommended for all pregnant women. However, SUC often misses microbes detected by expanded quantitative urine culture methods (EQUC) and 16S rRNA gene (amplicon) sequencing. The existence of these microbes (urinary microbiome) challenges the current approach to ASB screening in pregnancy as it is limited by two assumptions: (i) the female bladder is naturally sterile and (ii) SUC identifies all clinically relevant uropathogens. To determine the uniqueness or similarity between urinary and vaginal microbiomes, we performed a prospective observational institutional review board (IRB)-approved cohort study of pregnant women with a singleton pregnancy <14 weeks undergoing routine ASB screening. Consented subjects provided paired midstream voided urine and vaginal swab samples, which were assessed by EQUC and amplicon sequencing. The similarity was determined by Spearman's correlation; statistical significance was calculated using bootstrap analysis with 1,000 random samples and a significance threshold of $P$ value < 0.001. We used the Bayes theorem to quantify how well the vaginal microbiome could be used as a proxy for a patient's urinary microbiome and vice versa. Our findings provide evidence that EQUC and amplicon sequencing reveal similar views of urinary and vaginal microbiomes in first trimester pregnant women. However, while vaginal and urinary microbial compositions were significantly correlated for the same individual, they were by no means equivalent. The first trimester urinary and vaginal microbiomes are distinct enough to preclude their use as proxies of each other.

**IMPORTANCE** Untreated asymptomatic bacteriuria (ASB) has been associated with adverse pregnancy outcomes, including pyelonephritis, preterm labor, and low birth weight infants. Thus, routine screening by standard urine culture (SUC) and treatment of ASB are currently recommended for all pregnant women. For this purpose, some researchers claim that vaginal swabs and urine samples can be used as proxies for each other. Because SUC often misses microbes, we used two more sensitive, recently validated detection methods to compare the composition of the urinary and vaginal microbiomes of pregnant females in their first trimester. Both methods yielded similar results. Vaginal and urinary microbial compositions for the same individual were significantly correlated; however, they were not equivalent. We argue that first trimester urinary and vaginal microbiomes are distinct enough to preclude their use as proxies for each other.

**KEYWORDS** asymptomatic bacteriuria, enhanced culture, microbiome, pregnancy, 16S rRNA gene sequencing, clinical microbiology

I n the general obstetric population, 2%–11% of gravid women have evidence of uropathogenic bacteria without relevant signs or symptoms of a urinary tract infection

Address correspondence to Alan J. Wolfe, awolfe@luc.edu.

A.J.W. discloses research support from NIH, the Neilsen Foundation, and Pathnostics. He also discloses membership on the advisory boards of Urobiome Therapeutics and Pathnostics. The remaining authors report no disclosures.

See the funding table on p. 12.

(UTI) at the time of routine screening; this is referred to as asymptomatic bacteriuria (ASB) (1). Based on observational data regarding adverse pregnancy outcomes with ASB, screening and treatment of ASB during pregnancy are recommended for all pregnant women (2–4). However, the current approach to ASB screening is limited by two factors: (i) the assumption that the female bladder naturally exists in a sterile state and (ii) the assumption that all clinically relevant uropathogens are reliably identified by standard urine culture. Recent data from the literature on non-pregnant women have challenged both these assumptions (5–8).

It is now clear that the lower urinary tract of non-pregnant women contains a community of microbes, called the female urinary microbiome or urobiome. The urobiome was discovered using culture-independent DNA-based approaches. Using 16S rRNA gene sequencing, multiple groups identified bacterial DNA in urine collected either by using a transurethral catheter or by the so-called "clean catch" midstream voided methods from symptomatic and asymptomatic non-pregnant women (9–15). To validate these findings and to determine whether the sequenced bacteria were alive, several enhanced urine culture techniques were developed (16–18). These methods vastly outperform traditional urine culture methods in detecting bacteria, including species relevant to ASB (19).

Evidence now exists that the lower urinary tracts of pregnant women also contain microbes. Two teams of researchers have used 16S rRNA gene sequencing and/or enhanced culture methods to detect diverse microbes in midstream voided urine samples collected from women in the second trimester (20) and in urine obtained by transurethral catheterization from women who presented for delivery (21). However, these studies were limited by several factors, including assessment for the urobiome after the patient had already completed screening for ASB as part of their prenatal care. Therefore, we sought to evaluate and characterize the urobiome in the first trimester of pregnancy at the time of routine ASB screening. Furthermore, it is also unclear how much the urobiome reflects the vaginal microbiome or if the urobiome is distinct from the vaginal microbiome. As such, we compared paired voided urine and vaginal swab samples to determine the uniqueness or similarity between the urinary and vaginal microbiomes. To make this comparison, we used two complementary orthogonal methods: expanded quantitative urine culture (EQUC) and 16S ribosomal RNA gene amplicon sequencing of isolated DNA.

## MATERIALS AND METHODS

### Cohort selection

After receiving approval from the Loyola Institutional Review board (LU#209864), women who presented for their initial prenatal care appointment were screened for participation and recruited for a pilot study. We included pregnant women if they were English-speaking, ≥18 years old, in their first trimester (<14 weeks gestation) of a singleton pregnancy confirmed by ultrasound, and undergoing routine pelvic examination during their first prenatal visit. We excluded patients with a history of recurrent UTIs, antibiotic usage since conception, need for ongoing antibiotics (e.g., due to a history of recurrent UTI), multi-fetal gestations, gestation greater than 13 weeks and 6 days, urinary tract anomalies, or planning to deliver at an institution other than Loyola. Eligible participants provided written informed consent. At the time of enrollment, participants provided demographic information, medication use, and obstetrical history.

### Patient sample collection

All patient samples were collected as part of routine care for ASB screening prior to performing an examination. A midstream voided urine sample (described hereafter as "urine" samples) was collected in a sterile blue cap collection cup (BD #364956; Becton, Dickinson and Company). A portion of this sample was placed in a BD Vacutainer

Plus C&S Preservative Tube (Becton, Dickinson and Company), as is standard practice for clinical microbiology samples. At the time of the routine pelvic examination prior to performing a sterile digital examination, a vaginal sample (described hereafter as "vaginal" sample types) was collected with a sterile speculum from the posterior fornix of the vagina using the BD ESwab (BD#220245; Becton, Dickinson and Company, Sparks, Maryland), which contains a swab resting in 1 mL of the bacterial preservative. The microbiomes of all collected samples were determined using two methods: EQUC and 16S rRNA gene amplicon sequencing (described hereafter as the "amplicon" method).

## Urobiome identification by the EQUC method

Both the vaginal swab and voided urine samples were sent to the research laboratory for EQUC, which was initiated within 4 hours of collection, as previously described (17) (see also Table S1). Compared to the standard urine culture method, EQUC uses 100 times more urine, more types of media, and more atmospheric conditions with a longer incubation period. Thus, the detection limit for EQUC is 1 colony of growth on any plate or 10 colony forming units per milliliter (CFU/mL); in contrast, the detection limit for the standard method is 1,000 CFU/mL. If EQUC detected no bacteria, the sample was considered to be below the detection threshold. For each morphologically distinct colony type, CFU/mL were counted, and representative colonies were purified for identification by matrix-assisted laser desorption/ionization time-of-flight (MALDI-TOF) mass spectroscopy, as previously described (17).

## Microbiome identification by 16S amplicon sequencing

16S rRNA gene sequencing amplifies the V4 variable region of bacterial ribosomal RNA to identify live and dead bacteria. Genomic DNA (gDNA) was extracted from urine and vaginal samples using a Qiagen DNeasy Blood and Tissue kit via validated protocols (22). Controls were utilized in all steps to monitor for contamination. Gel electrophoresis and fluorescence dsDNA assays were used to monitor for DNA quality and quantity. gDNA was stored at −80°C until sequencing. 16S rRNA gene sequencing was performed on all available samples. The 16S rRNA gene was PCR-amplified from gDNA using degenerate primers (23) with index sequences and sequenced in pools on an Ilumina MiSeq at the Loyola Genomics Facility (24). Given that the urinary tract is a low biomass system and that 16S rRNA gene sequencing is highly sensitive, any contamination of the working space or sample may lead to skewed results. To limit false positives, controls were routinely utilized during processing. Furthermore, all samples were processed in duplicate to ensure reproducibility, with any discrepancy resulting in the processing of a third sample. Discrepancy was defined using the Bray–Curtis Dissimilarity Index, where a value of 0.5 was considered sufficiently different to merit a third sample.

Quality control and de-multiplexing of sequence data were performed with the onboard MiSeq Control software and MiSeq Reporter (current version: 2.1.43). The *mothur* pipeline was used to combine paired-end reads and remove contigs of incorrect length, contigs containing ambiguous bases, and chimeric sequences. Within *mothur,* sequences were assigned to operational taxonomic units based on a 97% similarity cutoff. Due to the low biomass nature of urine, the threshold for sequence positivity was set at a conservative cutoff of 2,000 sequence reads.

## Quantitative microbiome analyses

Prior to analysis, all EQUC- and amplicon-detected microbiomes were normalized to sum to an arbitrary value of 1,000. To determine how similar or dissimilar normalized urinary and vaginal microbiomes are to one another for the same subject and across all subjects, we used Spearman's correlation, a non-parametric test for assessing monotonic relationships between variables (25). Spearman's correlation ranges from −1 to 1, with 1 indicating perfect correlation, −1 indicating a perfect negative correlation, and 0 indicating no correlation. To consider correlations, we used the standard, if arbitrary,

ranges 0–0.19 as "very weak," 0.2–0.39 as "weak," 0.40–0.59 as "moderate," 0.6–0.79 as "strong," and 0.8–1 as "very strong." Statistical significance of Spearman's correlations was calculated using a bootstrap analysis with 1,000 random samples and a significance threshold of $P$ value < 0.001.

To quantify how well a vaginal microbiome could be used as a proxy for a patient's urobiome and vice versa, we used the following: for a specific patient, if a taxon is detected in one sample type (i.e., either vagina or urine), what is the conditional probability that it also will be detected in the other sample type? The answer was calculated using the Bayes theorem:

$$P_t(A\,|\,B) = \frac{P_t(B\,|\,A)P_t(A)}{P_t(B)} \, . \qquad (1)$$

where $P_t(A|B)$ is the probability that for taxon "$t$," if that taxon is found in sample type $B$, what is the probability that taxon $t$ will also be found in condition $A$? $P_t(A)$ is the probability that taxon t is in sample type $A$, and $P_t(B)$ is the probability that taxon $t$ is in sample type $B$. Only the presence or absence, and not relative abundance, was considered. To avoid calculating the probability for genera that are at the limits of detection, where presence and absence of a taxa may be attributable to chance, we eliminated the following from this analysis (i) the lowest abundant genera, accounting for the lowest 0.5% of abundance in observed microbiomes and (ii) genera that were detected in fewer than five samples.

## RESULTS

A total of 29 participants were recruited for the study. The population reflected the racially diverse obstetrics population at our institution. Table 1 displays the cohort's demographic characteristics and distribution.

Figures 1 and 2 present the urinary and vaginal microbiome composition at the genus level as detected by the EQUC and amplicon methods, respectively. The amplicon method detected 428 distinct genera, whereas the EQUC method detected 35 genera; 25 genera were detected by both methods (Fig. 3). Most (71%) of the genera detected

TABLE 1 Patient demographics and clinical characteristics[b]

| Parameter | Value ($n$ = 29 total patients) |
|---|---|
| Maternal age (years[a]) | 30 (19–42) |
| GA at recruitment[a] | 9w2d (4w6d-13w3d) |
| AMA age >35 yo | 3 (10.3%) |
| Nulliparity | 5 (17.2%) |
| Multiparity | 24 (82.8%) |
| BMI[a] | 26.61 (18–42.87) |
| Race | |
| Non-Hispanic White | 10 (34.5%) |
| African American | 7 (24.1%) |
| Hispanic | 8 (27.6%) |
| Asian | 2 (6.9%) |
| Others | 2 (6.9%) |
| Medical comorbidities | |
| Asthma | 3 (10.3%) |
| Gestational diabetes | 2 (6.9%) |
| History of pre-eclampsia | 2 (6.9%) |
| History of DVT | 2 (6.9%) |
| Thyroid | 4 (13.8%) |

[a]Median with range.
[b]GA: gestational age; AMA: advanced maternal age; BMI: body mass index; DVT: deep venous thrombosis.

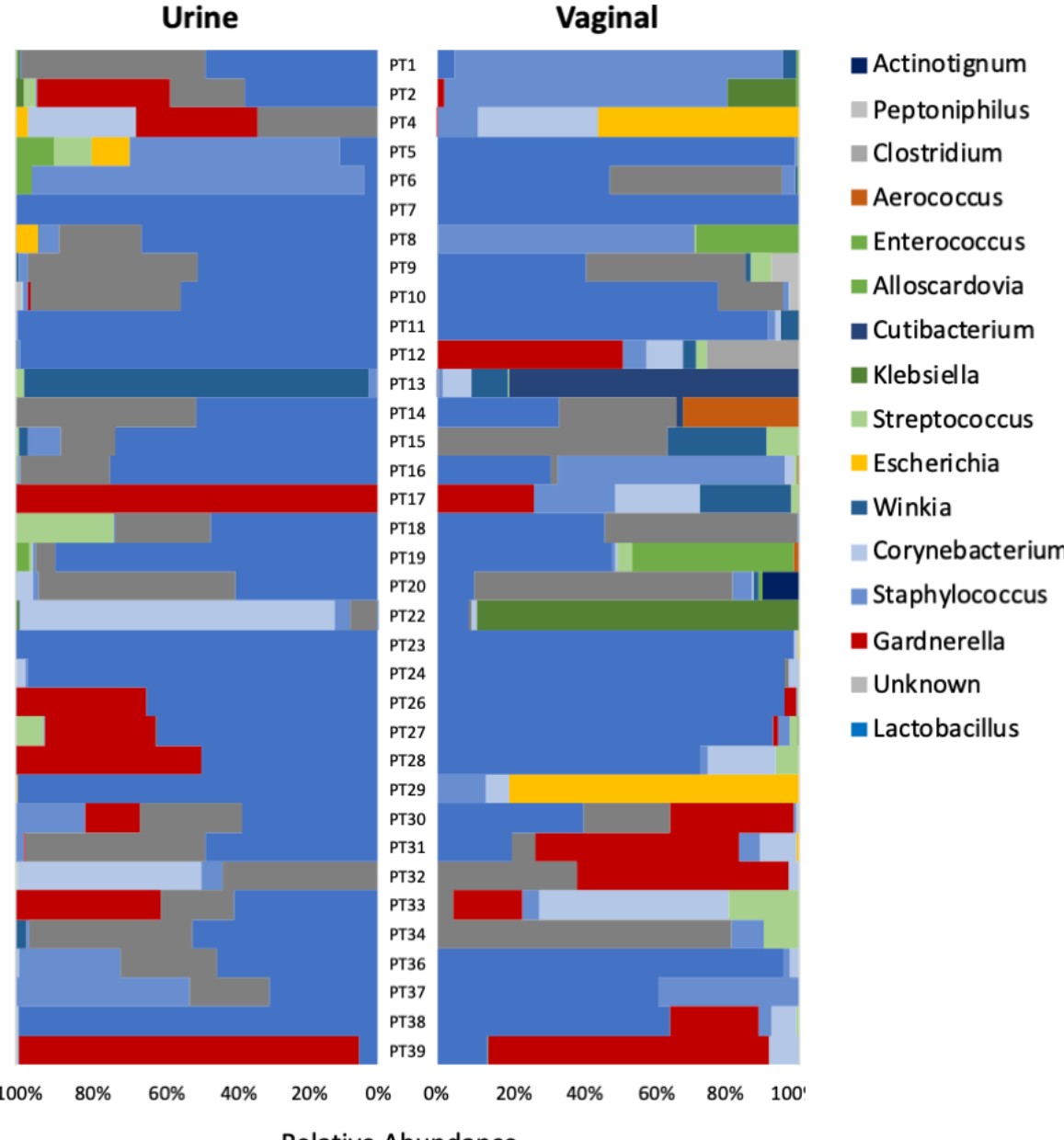

**FIG 1** Relative abundance of genera in urine and vaginal microbiomes by EQUC. Butterfly plot with the 95% most abundant taxa mapped to color and each microbiome community summing to 100%.

by EQUC were also detected by the amplicon method. Although the vast majority (94%) of genera detected by the amplicon method were undetected by EQUC, the 25 commonly detected genera were the most abundant, accounting for 82% and 85% of total abundance by the EQUC and amplicon methods, respectively.

Sample types and methods can be compared by considering the average of all urobiomes for each sample type and analysis method (Fig. 3). By both methods, on average, the genus *Lactobacillus* predominated in both microbiomes; however, *Lactobacillus* occupied a higher proportion of the urobiome as observed by the amplicon method than by EQUC. *Gardnerella* was the second most abundant genus in both sample types and by both methods.

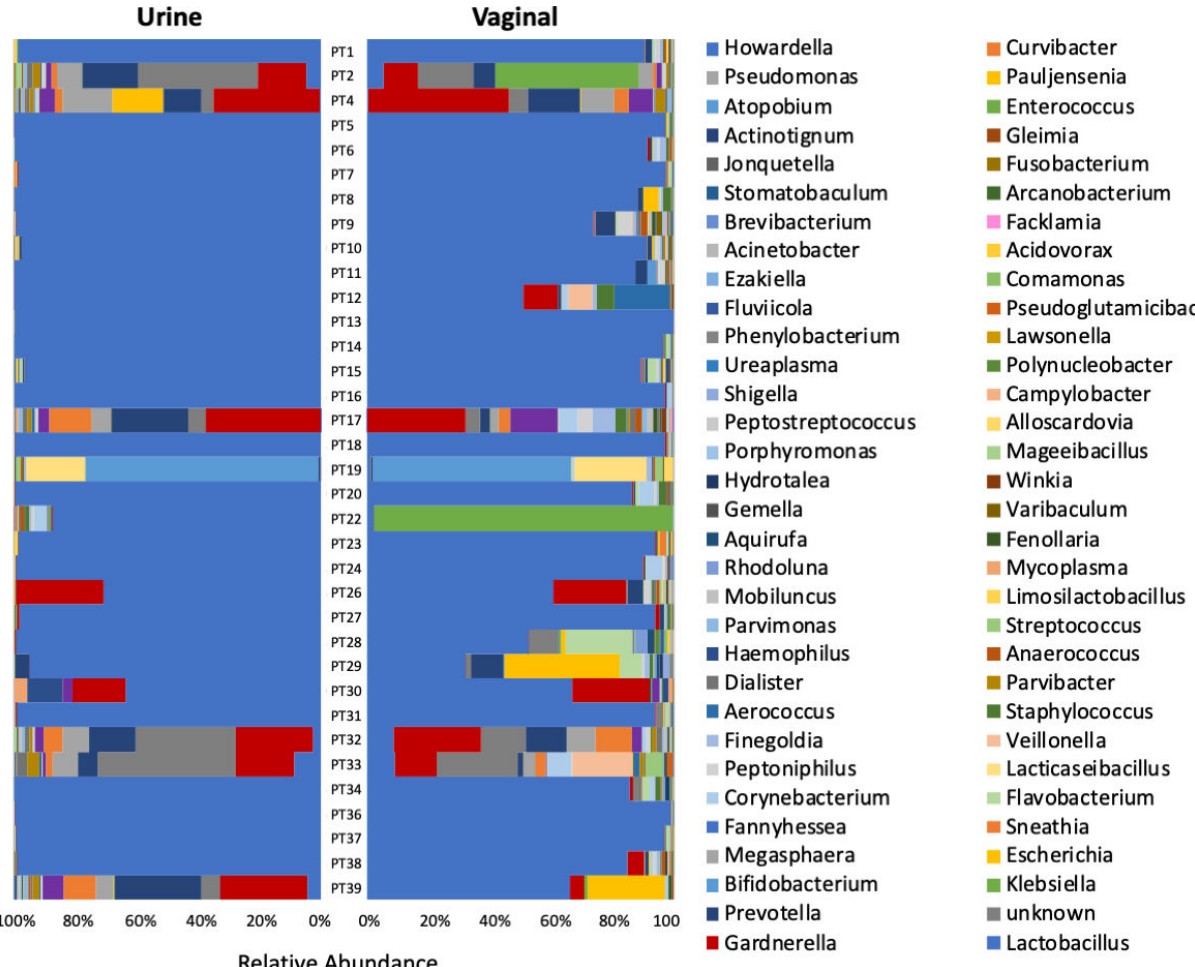

**FIG 2** Relative abundance of genera in urine and vaginal microbiomes by 16S rRNA gene sequencing. Butterfly plot with the 95% most abundant taxa mapped to color and each microbiome community summing to 100%.

## How similar are the microbiomes as detected by the amplicon and EQUC methods?

We next compared the two different detection methods, asking how well they agreed with each other when used to analyze the same microbiome community. Considering the genera detected by both methods, there was a "moderate" and statistically significant correlation between the methods for the same sample, relative to the correlation between other samples of the same sample type (Spearman's correlation of 0.48 for vagina and 0.53 for urine sample types. Bootstrap *P* values less than 0.001). However, when considering the average abundances of these 25 commonly detected genera, the alpha diversity of the microbiomes, as calculated by the Shannon Diversity Index, differed substantially by the analytic method. The microbiomes detected by EQUC (Shannon's Diversity Index of 1.6 and 1.2 for vaginal and urine microbiomes, respectively) were more diverse than those detected by the amplicon method (0.5 and 0.9 for vaginal and urine microbiomes, respectively). This difference in diversity is due to the predominance of *Lactobacillus* in amplicon-detected microbiomes relative to EQUC-detected microbiomes.

## How similar are the urine and vaginal microbiomes?

To determine if the microbiome of one sample type is a good proxy for the other, we compared the paired vaginal and urine microbiomes from each participant. For the

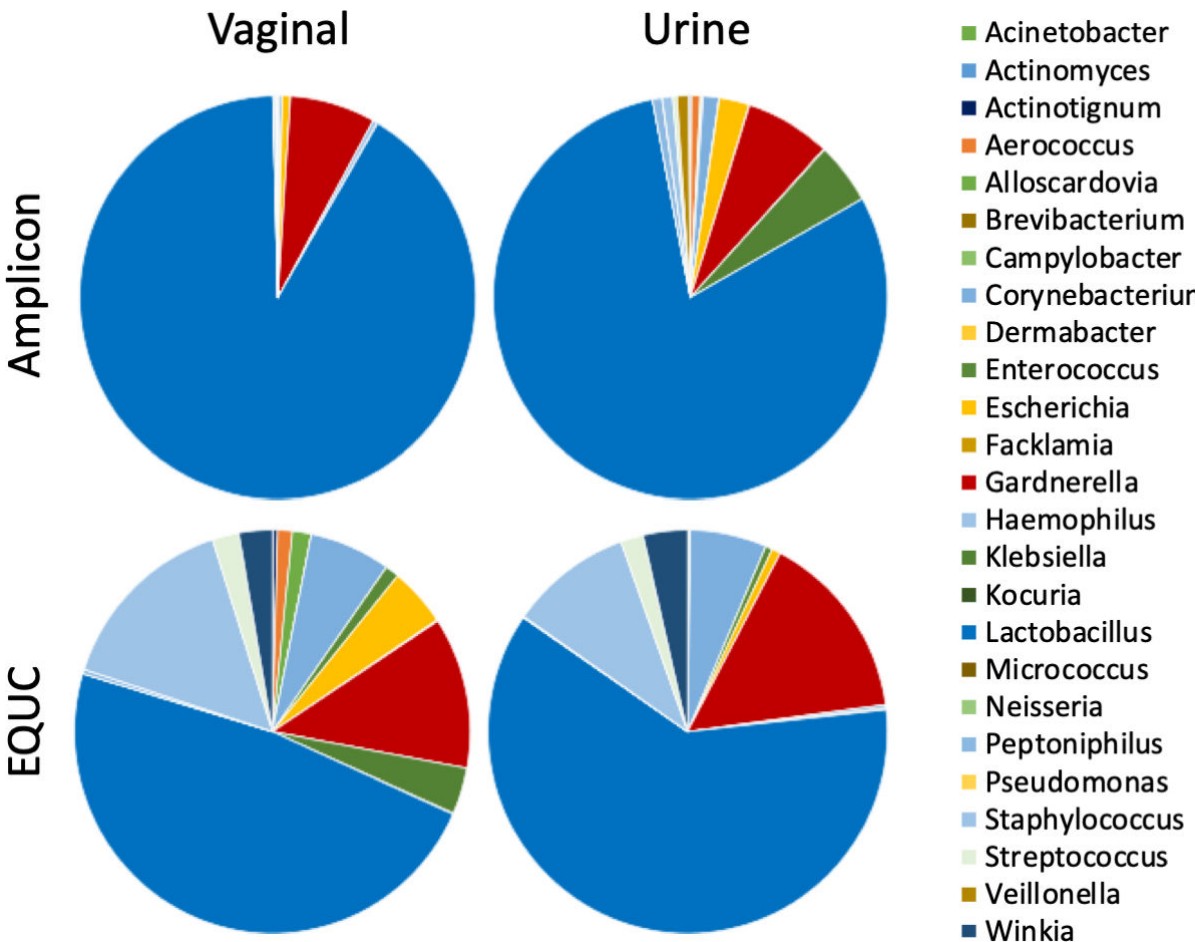

**FIG 3** Average abundance of taxa in swab and urine communities by culture and amplicon methods. The 25 genera in common between amplicon and culture methods are shown in pie-charts.

microbiomes as detected by the amplicon method, there was a "strong" (Spearman's correlation coefficient = 0.60) and statistically significant (Bootstrap x1000, pVal <0.001) correlation between vaginal and urine microbiomes of the same subject. For EQUC-detected microbiomes, there also was a "strong" (Spearman's correlation coefficient = 0.66) and statistically significant (Bootstrap x1000, pVal <0.001) correlation between vaginal and urine microbiomes of the same subject. Furthermore, while not identical, the vaginal and urine microbiomes were more similar to one another for the same subject than they were between subjects.

Given the observation that vaginal and urine sample types were broadly and significantly similar for a given subject, we could now address the question: if a genus was detected in one sample type, what was the likelihood that the genus was detected in the other? Thus, we calculated the conditional probability that a genus was present in one sample type, given that it had been detected in the other (Tables 2 and 3). As they have no predictive power, we eliminated from the analysis very low abundance genera and those that occurred infrequently. For the vaginal and urine microbiomes, this removed 11 of 35 (31%) genera for the EQUC-detected microbiomes and 64 of 428 (15%) genera for the amplicon-detected microbiomes.

**How good is the vaginal microbiome at predicting the urobiome?**

In EQUC-detected microbiomes, for 36% of the detected genera (4 of 11), if a genus was detected in the vaginal microbiome, there was at least a 75% chance of also being detected in the urobiome. These five genera constituted 79% of the total abundance in

**TABLE 2** Conditional probability of detecting taxa in EQUC-analyzed samples[c]

| Swab genus | P (vaginal\|urine)[a] | P (urine\|vaginal)[b] |
|---|---|---|
| *Lactobacillus* | **0.813** | **1.000** |
| *Staphylococcus* | **0.867** | **0.867** |
| Unknown | 0.636 | **0.824** |
| *Gardnerella* | **0.818** | 0.750 |
| *Candida* | 0.600 | 0.750 |
| *Klebsiella* | 0.667 | 0.667 |
| *Corynebacterium* | **0.929** | 0.591 |
| *Escherichia* | 0.667 | 0.571 |
| *Winkia* | 0.500 | 0.556 |
| *Streptococcus* | **0.846** | 0.524 |
| *Haemophilus* | 0.000 | Urine only |

[a]The probability that genera will be detected in the vaginal swab if it is detected in the urine sample.
[b]The probability that genera will be detected in the urine sample swab if it is detected in the vaginal swab.
[c]Bold values are greater than 0.75.

the urobiome. There was no correlation ($R^2 = 0.07$) between abundance in the urobiome and $P_t$(urine\|vaginal) (i.e., the probability of observing taxon "*t*" in the urine microbiome given that it was observed in the vaginal microbiome).

In amplicon-detected urobiomes, for 19% of the detected genera (16 of 64), if a genus was found in the vaginal microbiome, there was at least a 75% chance of it also being detected in the urobiome. These 16 genera constituted 77% of the total abundance of the urobiome. There was no correlation ($R^2 = 0.06$) between abundance in the urobiome and $P_t$(urine\|vaginal).

## How good is urine at predicting the vaginal microbiome?

In EQUC-detected microbiomes, for 45% of the detected genera (five of 11), if a genus was detected in the urobiome, there was at least a 75% chance of also being present in the vaginal microbiome. These five genera constituted 79% of the total abundance in the vaginal microbiome. There was a "moderate" correlation ($R^2 = 0.75$) between abundance in the urobiome and $P_t$(vaginal\| urine) (i.e., the probability of observing taxon "*t*" in the vaginal microbiome given that it was observed in urobiome), indicating that those genera in the urobiome that were predictive for the vaginal microbiome tend to be from the higher-abundance genera.

In amplicon-detected microbiomes, for 10% of the detected genera (eight of 64), if a genus was found in the urobiome, there was at least a 75% chance of also being detected in the vaginal microbiome. These eight genera constituted 82% of the abundance in the vaginal microbiome. There was no correlation ($R^2 = 0.09$) between abundance in the urobiome and $P_t$(vaginal\|urine).

Not all genera are equivalent when the goal is to diagnose ASB in patients. The taxa considered significant for ASB diagnosis include *Escherichia coli, Enterococcus* species (i.e., *E. faecalis* and *E. faecium*), *Klebsiella* species (especially *K. pneumoniae*), coagulase-negative *Staphylococcus* species (especially *S. saprophyticus*), *Pseudomonas,* and Group B *Streptococcus* (i.e., *S. agalactiae*) (26, 27). The set of conditional probabilities for these and other urobiome genera is presented in Tables 2 and 3 for the EQUC and amplicon microbiome methods, respectively. EQUC detected *Staphylococcus*, *Klebsiella*, *Escherichia,* and *Streptococus,* as well as the yeast pathogen *Candida* and several other genera, including *Gardnerella*, which is associated with bacterial vaginosis. With the exception of *Candida*, the amplicon method detected these and many more genera.

For EQUC-detected microbiomes, if *Staphylococcus* was found in the urine microbiome, then then there was at least a 75% chance that it would also be found in the vaginal microbiome, and vice versa. This was not true for all ASB genera, however. For example, if *Streptococcus* was found in the urine microbiome, then then there was at least a 75% chance it would also be found in the vaginal microbiome, but not vice versa.

**TABLE 3** Conditional probability of detecting taxa in 16S amplicon-method samples[c]

| Amplicon genus | P (vaginal\|urine)[a] | P (urine\|vaginal)[b] |
|---|---|---|
| *Acidovorax* | 0.056 | 0.200 |
| *Acinetobacter* | 0.000 | 0.000 |
| *Actinotignum* | 0.077 | 0.333 |
| *Aerococcus* | 0.444 | 0.571 |
| *Alloscardovia* | 0.167 | **1.000** |
| *Anaerococcus* | 0.548 | 0.654 |
| *Aquirufa* | 0.185 | 0.455 |
| *Arcanobacterium* | 0.500 | 0.667 |
| *Atopobium* | 0.083 | 0.167 |
| *Bifidobacterium* | 0.278 | 0.227 |
| *Brevibacterium* | 0.200 | 0.167 |
| *Campylobacter* | 0.160 | 0.444 |
| *Comamonas* | 0.105 | 0.667 |
| *Corynebacterium* | 0.727 | **0.800** |
| *Curvibacter* | 0.063 | 0.333 |
| *Dialister* | 0.423 | 0.579 |
| *Enterococcus* | 0.133 | 0.333 |
| *Escherichia* | 0.706 | **0.828** |
| *Ezakiella* | 0.417 | 0.455 |
| *Facklamia* | 0.059 | 0.143 |
| *Fannyhessea* | 0.500 | 0.500 |
| *Fenollaria* | 0.381 | 0.571 |
| *Finegoldia* | 0.594 | **0.792** |
| *Flavobacterium* | **0.818** | **0.871** |
| *Fluviicola* | 0.091 | 0.500 |
| *Fusobacterium* | 0.154 | 0.286 |
| *Gardnerella* | 0.563 | 0.643 |
| *Gemella* | 0.625 | 0.500 |
| *Haemophilus* | 0.333 | 0.400 |
| *Howardella* | 0.286 | 0.500 |
| *Hydrotalea* | 0.111 | 0.375 |
| *Jonquetella* | 0.167 | 0.250 |
| *Klebsiella* | 0.563 | 0.750 |
| *Lacticaseibacillus* | **1.000** | **1.000** |
| *Lactobacillus* | **1.000** | **1.000** |
| *Lawsonella* | 0.208 | 0.500 |
| *Leucobacter* | 0.000 | Urine only |
| *Levyella* | 0.083 | 0.250 |
| *Limosilactobacillus* | **0.923** | **0.923** |
| *Mageeibacillus* | 0.750 | **1.000** |
| *Megasphaera* | 0.500 | 0.500 |
| *Mobiluncus* | 0.467 | 0.636 |
| *Mycoplasma* | **1.000** | 0.714 |
| *Parvibacter* | **0.833** | **0.833** |
| *Parvimonas* | 0.625 | 0.556 |
| *Pauljensenia* | **1.000** | **1.000** |
| *Peptoniphilus* | **0.767** | **0.793** |
| *Peptostreptococcus* | 0.267 | 0.571 |
| *Phenylobacterium* | 0.074 | 0.333 |
| *Polynucleobacter* | 0.125 | 0.500 |
| *Porphyromonas* | 0.250 | 0.313 |
| *Prevotella* | 0.727 | **0.828** |

**TABLE 3** Conditional probability of detecting taxa in 16S amplicon-method samples[c] (*Continued*)

| Amplicon genus | P (vaginal|urine)[a] | P (urine|vaginal)[b] |
|---|---|---|
| *Pseudoglutamicibacter* | 0.000 | 0.000 |
| *Pseudomonas* | 0.056 | 0.500 |
| *Rhodoluna* | 0.222 | 0.667 |
| *Shigella* | 0.200 | 0.500 |
| *Sneathia* | 0.500 | 0.455 |
| *Staphylococcus* | 0.576 | **0.864** |
| *Streptococcus* | 0.273 | 0.500 |
| Unknown | 0.742 | **0.821** |
| *Ureaplasma* | 0.583 | 0.583 |
| *Varibaculum* | 0.208 | 0.313 |
| *Veillonella* | 0.250 | 0.250 |
| *Winkia* | 0.250 | 0.417 |

[a]The probability that genera will be detected in the vaginal swab if it is detected in the urine sample.
[b]The probability that genera will be detected in the urine sample swab if it is detected in the vaginal swab.
[c]Bold values are greater than 0.75.

In contrast. the presence of *Klebsiella* was completely uninformative; if *Klebsiella* was found in either the vaginal or urine microbiomes, there was not a good chance (i.e., less than 75%) that it would also be found in the other sample type.

For amplicon-detected microbiomes, if an ASB-relevant genus was found in one sample type, there was often not a good chance (i.e., less than 75%) that it would be found in the other sample type. For example, if *Staphylococcus* or *Escherichia* was found in the vaginal microbiome, then there was at least a 75% chance that the genus would also be found in the urine microbiome, but not vice versa. Again, the presence of *Klebsiella* was completely uninformative.

## DISCUSSION

### Principal findings

Enhanced culture (e.g., EQUC) and amplicon-based (e.g., 16S rRNA gene sequencing) methods are orthogonal and complementary approaches to assess urogenital microbiomes (i.e., voided urine and vagina). The first trimester urinary and vaginal microbiomes are distinct enough to preclude their use as proxies of each other.

### Results

The EQUC and amplicon methods each provide unique insights and biases into the analysis of microbiomes. Enhanced culture methods, such as EQUC, identify living cells but cannot identify bacteria that cannot be cultured using the selected experimental conditions. Amplicon-based methods, such as 16S RNA variable-region analysis, cannot distinguish between living and dead cells or even cell-free DNA but can detect non-culturable bacteria. Here, we have shown that these two methods arrive at microbiome compositions that moderately but significantly correlate, indicating that both methods uncover the same microbiome communities, albeit with method-specific biases.

Vaginal and urinary microbiomes are, for the same patient, significantly correlated by their compositions, due primarily to the more relatively abundant genera. However, vaginal and urinary microbiomes are by no means equivalent. For only a minority of genera, if a genus was detected in one sample type, there was at least a 75% chance that that genus would be present in the other sample type.

### Clinical implications

The emerging evidence that the lower urinary tracts of pregnant women are not sterile and that the vaginal and lower urinary tract microbiomes are often dissimilar complicates ASB assessment and therefore treatment. Thus, during ASB assessment, we

cannot assume that information derived from vaginal swabs can be extrapolated to the microbes of the lower urinary tract. A larger number of studies to extrapolate to clinical correlations are warranted.

## Strengths and limitations

Our study has limitations. Our findings are limited by our small sample size of the study and the use of midstream voided urine instead of catheterized urine. The vaginal swabs were collected by different providers, and the location of posterior fornix from which the sample was obtained may have varied. There was no patient follow-up to link microbiome composition to ASB diagnosis. The lack of longitudinal samples precluded opportunities to determine if the observed urinary and vaginal microbiomes are characteristic for each patient or dynamic over time.

This study also has strengths. We compared paired vaginal and urine samples collected simultaneously. We used orthogonal methods of microbiome analysis; both led to similar conclusions, providing confidence that our observations were a function of biological phenomenon and not due to technical biases.

## Conclusion

Our findings provide evidence that EQUC and 16S rRNA gene sequencing are complementary methods that reveal similar views of the urinary and vaginal microbiomes of first trimester pregnant women. Our findings also show that urine samples and vaginal swabs sample related microbiological niches; however, those niches are distinct enough that they should not be used as proxies for each other.

## ACKNOWLEDGMENTS

We wish to acknowledge Mary Tulke for her help in obtaining IRB approval and patient recruitment, the Loyola Genomics Facility for 16S rRNA gene sequencing, members of the Wolfe laboratory for processing samples and performing EQUC, and our study participants.

We acknowledge the following funding: NIH R01DK104718 awarded to A.J.W. The funding source had no role in study design, collection, analysis, and interpretation of data, in the writing of the report, or the decision to submit the article for publication.

A.J.W. obtained funding. J.S., T.P.W., and A.J.W. designed the study. J.S. recruited participants. T.M.H. processed the specimens. T.M.H. and P.L. analyzed the data. J.S., P.L., and A.J.W. wrote the manuscript. T.P.W., J.R.G., and A.J.W. provided oversight. All authors reviewed and edited the manuscript.

## AUTHOR AFFILIATIONS

[1]Department of Obstetrics and Gynecology, Division of Maternal Fetal Medicine, Loyola University Medical Center, Maywood, Illinois, USA
[2]Loyola Genomics Facility, Stritch School of Medicine, Loyola University Chicago, Chicago, Illinois, USA
[3]Department of Microbiology and Immunology, Stritch School of Medicine, Loyola University Chicago, Chicago, Illinois, USA

## PRESENT ADDRESS

Juliana Sung, Department of Obstetrics and Gynecology, Division of Maternal Fetal Medicine, Loyola University Medical Center, Maywood, Illinois, USA
Thaddeus P. Waters, Department of Obstetrics & Gynecology, Jacobs School of Medicine and Biomedical Sciences, University at Buffalo, Buffalo, New York, USA
Jean R. Goodman, Department of Obstetrics, Gynecology and Women's Health, University of Missouri-Columbia, Columbia, Missouri, USA

## AUTHOR ORCIDs

Alan J. Wolfe  http://orcid.org/0000-0003-4532-0545

## FUNDING

| Funder | Grant(s) | Author(s) |
|---|---|---|
| HHS | NIH | National Institute of Diabetes and Digestive and Kidney Diseases (NIDDK) | R01DK104718 | Alan J. Wolfe |

## AUTHOR CONTRIBUTIONS

Juliana Sung, Conceptualization, Data curation, Methodology, Writing – original draft, Writing – review and editing | Peter Larsen, Formal analysis, Visualization, Writing – original draft, Writing – review and editing | Thomas M. Halverson, Formal analysis, Investigation, Methodology, Writing – review and editing | Thaddeus P. Waters, Conceptualization, Supervision, Writing – review and editing | Jean R. Goodman, Supervision, Writing – review and editing | Alan J. Wolfe, Conceptualization, Funding acquisition, Project administration, Resources, Supervision, Writing – original draft, Writing – review and editing

## DATA AVAILABILITY

The raw sequencing files are available in the NCBI under BioProject accession no. PRJNA1034602.

## ADDITIONAL FILES

The following material is available online.

### Supplemental Material

**Table S1 (Spectrum02638-23-s0001.docx).** Culture protocols: SUC versus EQUC.

### Open Peer Review

**PEER REVIEW HISTORY (review-history.pdf).** An accounting of the reviewer comments and feedback.

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
