## [Reviewer comments · Microbiology Spectrum]

Microbiology Spectrum

First trimester ‘clean catch’ urine and vaginal swabs sample distinct microbiological niches

Juliana Sung, Peter Larsen, Thomas Halverson, Thaddeus Waters, Jean Goodman, and Alan Wolfe

Corresponding Author(s): Alan Wolfe, Loyola University Chicago - Health Sciences Campus

Review Timeline:

Submission Date:	June 26, 2023
Editorial Decision:	September 21, 2023
Revision Received:	October 16, 2023
Accepted:	November 20, 2023

Editor: Kevin Theis

Reviewer(s): Disclosure of reviewer identity is with reference to reviewer comments included in decision letter(s). The following individuals involved in review of your submission have agreed to reveal their identity: Jonathan M Greenberg (Reviewer #1); Luísa Peixe (Reviewer #2)

Transaction Report:

DOI: <https://doi.org/10.1128/spectrum.02638-23>

September 21, 2023

Dr. Alan J Wolfe
Loyola University Chicago - Health Sciences Campus
Microbiology and Immunology
Maywood, IL

Re: Spectrum02638-23 (First trimester 'clean catch' urine and vaginal swabs sample distinct microbiological niches)

Dear Dr. Alan J Wolfe:

Thank you for submitting your manuscript to Microbiology Spectrum, and thank you for your patience during the review process. The manuscript has been reviewed by two experts in the field. Although both are appreciative of the data, some modifications are required for publication. The comments of the reviewers are detailed below. Please email or call me if you have any questions.

Link Not Available

Sincerely,

Kevin Theis

Journals Department
Reviewer comments:

Reviewer #1 (Comments for the Author):

The manuscript titled "First trimester 'clean catch' urine and vaginal swabs sample distinct microbiological niches" aims to evaluate the urobiome (urinary microbiome) in comparison to the vaginal microbiome and assess whether vaginal samples are efficient enough to characterize the urobiome in pregnant women.

The general approach of the study was to evaluate similarities between the vaginal and urinary microbiomes of women during the 1st trimester of pregnancy using two different approaches and to evaluate how predictive one sample type would be for the other.

The key findings of the study were that the composition of paired vaginal and urinary microbiomes is similar, but with differences in lesser relatively abundant taxa. Additionally, there was a moderate correlation between the microbiomes analysis methods via Amplicon and EQUIC, however the amplicon method detected a much larger number of genera.

The conclusions of the study were that there is a moderate and statistically significant correlation between the two microbiome survey methods, with inherent biases of each method leading to minor differences in the compositions. More substantially, the urinary and vaginal microbiomes from the same patients, regardless of analysis method are similar yet not equivalent, and thus, clinically, their use as proxies for each other needs to be further evaluated due to the clinical impacts of ASB and current screening practices.

Comments/Concerns

This study utilizes an enhanced culture technique relative to standard clinical cultures and amplicon sequencing to establish congruence of the two methodologies. The authors provide strong evidence that EQUIC can capture the most relatively abundant members of the vaginal and urinary microbiomes as detected by amplicon sequencing, and importantly, a number of ASB-associated taxa. Furthermore, the authors establish that paired vaginal and urine samples are similar in composition within subject, but not effectively enough to consider them as fully representative of the alternative sample type.

This study provides valuable insights into the importance of ASB screening and either sample type alone, is likely not sufficient to assess the microbiomes of both sites, and subsequently assess ASB/UTI risk; however, I have several concerns with it as written:

A principal concern with the results as illustrated is the lack of clarity as to what cutoffs were used for the genera included in Figures 1 and 2. There are over 50 genera in the EQUIC figure and only 12 in the amplicon figure. Was a minimum relative abundance threshold applied for these figures? If so, please indicate. The second paragraph emphasizes that 25 genera were shared, perhaps including the 25 shared in these figures would add clarity. Minimally, please indicate what criteria were used for inclusion of genera in the figure legends.

Additionally, the figure quality (i.e. resolution) on all three primary figures is poor and makes some of the labels/names hard to read. I recommend updating them to higher resolution.

Lastly, I noticed several inconsistencies and spots of poor clarity in the results section and recommend the authors revisit this portion after considering the minor comments/edits indicated below.

Minor comments/edits:

Vagina and urine are intermittently capitalized throughout the manuscript, please correct for consistency.

Abstract

No minor comments.

Introduction

No minor comments.

Methods

Line 110: Correct spelling of "Institutional"

Line 117: Change "delivery" to "deliver"

Line 124: Delete "a sterile"

Line 132: Change semicolon to an apostrophe

Line 174: Correct spelling of "perfect"

Be consistent with the capitalization of "Correlation"

Line 183: Should each P be accompanied by a subscript "t"?

Line 184: Should "condition" be "Sample type"?

Results

Line 192: Spell out OB.

"Moderate" is indicated as a correlation range between 0.40-0.59 in the methods section, however in several instances, the correlations are equal to or greater than 0.60 and referred to as "Moderate". Please correct these instances.

Fix capitalization and quotations for consistency of "Moderate" also throughout the Results section.

Paragraph starting at Line 250: Please double-check the numbers, you state 8 of 64 genera then 16 genera in the subsequent sentence. The table the data is referencing indicate it should be 16 genera for this section.

Paragraphs at Line 243 and Line 258 appear to be copied with some numbers being incorrect. Please address and consider rewording the paragraphs in these sections so they are not so identical.

Line 283: Add "ASB-associated" before "genera".

Paragraph at Line 288: The first two sentences are in contradiction. You state there was never a good chance and then in the subsequent sentence two genera are identified that had a good chance to be present.

Figures 1 and 2: Urine and Vaginal labels are cut off at the top of both figures and the Relative abundance label in Figure 2.

Tables 2 and 3: Add clarification of what the $p(\text{vaginal}|\text{urine})$ and $p(\text{Urine}|\text{vaginal})$, i.e. probably of taxa detected in urine to also be present in the vaginal microbiome, in a note or indicate as each column label.

Discussion

In the results portion of the discussion, I suggest reiterating the finding that the more relatively abundant taxa were more likely to have high probability of to occur in the other sample type.

Reviewer #2 (Comments for the Author):

The manuscript authored by Wolfe et al. aims to compare the vaginal and urinary microbiomes of women during the first trimester of pregnancy, employing two distinct methodologies: EQUIC and 16S rRNA gene (amplicon) sequencing. While I find the authors' perspective on the ASB concept and its pertinence in light of contemporary insights into the urinary microbiome and agree with it entirely, the study fails in its ability to fully address this facet of the objectives. This deficiency arises from: 1) the study design supported on the genus analysis; 2) Microbiome characterization methodology; 3) omission of a follow-up on the participants concerning ASB diagnosis and potential therapeutic interventions; 4) data analysis performed.

Following several flaws are detailed.

The study design supported on the genus analysis is an important flaw, namely when it's known that for the most prevalent and abundant genus/family different species (e.g. Lactobacillaceae 14 species of 4 genera; Corynebacterium 25 species; Gardnerella 9 species/genomospecies; Staphylococcus 14 species; Streptococcus 10 species) have been described and with different pathogenic potential.

Furthermore, concerning the matter of methodology, the utilization of solely the V4 region of the 16S gene significantly undermines the capacity for precise bacterial identification, as it restricts the outcomes to genus-level categorization. The methodology used for the EQUIC also have limitations in capturing the full bacterial diversity and bacterial groups abundance. The plates used have low colony discriminatory ability for Gram positive and Gram negative bacterial genera. Moreover, the authors only characterized a distinct colony morphotype per plate. Additionally, the employment of MALDI-TOF MS identification also reveals limitations, primarily due to prevailing databases being predominantly populated by clinically significant pathogens, thereby neglecting a substantial portion of the microbial species within the microbiome. In tandem with the inherent disparities in sample processing between these two methods, it becomes foreseeable that any meaningful correlation between their findings might prove elusive.

In addition, regarding methodology, an important concern arises from the criteria used to calculate the likelihood of a particular genus being present in one sample and concurrently identified in the other. To achieve this, the authors opted to exclude genera with very low abundances from the analysis, specifically those accounting for the lowest 0.5% of the total abundance. Regrettably, this strategy led to the inclusion of a mere 11 out of 35 (31%) genera from the EQUIC-detected microbiomes and 64 out of 428 (15%) genera from the amplicon-detected microbiomes. This approach, in effect, injects a noticeable bias into the analysis by discarding a significant portion of the bacterial community that was originally detected. Given the intrinsic nature of the urinary urobiome, where diversity is pronounced and many genera are naturally present in minimal abundances, the choice to analyze only 31% of the genera obtained through EQUIC and a mere 15% of the community identified through amplicon sequencing stands as an erroneous probability of the presence of a given genus in one sample to be identified in the other. It is my opinion that this analysis should be repeated with the majority of the identified community.

Throughout the manuscript, there are instances of conflicting ideas regarding the correlation between the two microbiomes. These disparities should be highlighted to clarify the study's perspective and its alignment with prior research findings. For instance, in the abstract, a sentence states, "Bayes Theorem quantified how well the vaginal microbiome could be used as a proxy for a patient's urinary microbiome and vice versa." However, in the concluding paragraph, it is stated, "While vaginal and urinary microbial compositions were significantly correlated for the same individual, they were by no means equivalent. The first trimester urinary and vaginal microbiomes are distinct enough to preclude their use as proxies of each other."

Another conflicting notion is presented in the abstract: EQUIC and amplicon sequencing are portrayed as complementary, unveiling analogous perspectives of the urinary and vaginal microbiomes in first-trimester pregnant women. However, if these methodologies indeed yield similar viewpoints, their complementarity might be questioned. It could be argued that true complementarity arises when their findings necessitate consolidation. Alternatively, if one methodology proves more elucidating, a discerning selection could be made to favor one over the other.

Specific points

Material and Methods

Line 137- details about media suppliers are missing. This is important to check the medium composition, e.g. the difference between Aerobic Blood agar and blood agar.

Line 144- Details about the brand and model of Maldi-TOF MS equipment should be provided, as well as interpretation threshold used for considering the bacterial identification provided by the equipment.

Line 157-159 - which criteria was followed for considering 2 samples as discrepant?

The genus *Lactobacillus* is currently split into different genera. The authors should have used methodologies allowing Lactobacillaceae genus identification. The role of those genera and Lactobacillaceae species are being unveiled, suggesting different impacts in the microbiome of the urogenital tract.

"Lactobacillaceae" should replace "Lactobacillus" when mentioning several former *Lactobacillus* genera.

Results

Information on the number of colonies identified per plate should have been provided.

More detailed information per sample should have been provided (e.g. total bacterial cells count; counts for different genera; reads per genus).

Staff Comments:

Preparing Revision Guidelines

Please return the manuscript within 60 days; if you cannot complete the modification within this time period, please contact me. If you do not wish to modify the manuscript and prefer to submit it to another journal, please notify me of your decision immediately so that the manuscript may be formally withdrawn from consideration by Microbiology Spectrum.

Kevin,

We thank you and the reviewers for your/their efforts. As you will see below, we thought that reviewer #1 was dead on, pointing out some real issues with the manuscript. We believe that we have resolved each of those issues. In contrast, we thought the reviewer #2 was wrong on many of their concerns. We have tried to respectfully explain our point of view on each issue. We hope that our attempts are satisfactory.

Reviewer comments:

Reviewer #1 (Comments for the Author):

The manuscript titled "First trimester 'clean catch' urine and vaginal swabs sample distinct microbiological niches" aims to evaluate the urobiome (urinary microbiome) in comparison to the vaginal microbiome and assess whether vaginal samples are efficient enough to characterize the urobiome in pregnant women.

The general approach of the study was to evaluate similarities between the vaginal and urinary microbiomes of women during the 1st trimester of pregnancy using two different approaches and to evaluate how predictive one sample type would be for the other.

The key findings of the study were that the composition of paired vaginal and urinary microbiomes is similar, but with differences in lesser relatively abundant taxa. Additionally, there was a moderate correlation between the microbiomes analysis methods via Amplicon and EQUIC, however the amplicon method detected a much larger number of genera.

The conclusions of the study were that there is a moderate and statistically significant correlation between the two microbiome survey methods, with inherent biases of each method leading to minor differences in the compositions. More substantially, the urinary and vaginal microbiomes from the same patients, regardless of analysis method are similar yet not equivalent, and thus, clinically, their use as proxies for each other needs to be further evaluated due to the clinical impacts of ASB and current screening practices.

Comments/Concerns

This study utilizes an enhanced culture technique relative to standard clinical cultures and amplicon sequencing to establish congruence of the two methodologies. The authors provide strong evidence that EQUIC can capture the most relatively abundant members of the vaginal and urinary microbiomes as detected by amplicon sequencing, and importantly, a number of ASB-associated taxa. Furthermore, the authors establish that paired vaginal and urine samples are similar in composition within subject, but not effectively enough to

consider them as fully representative of the alternative sample type.

This study provides valuable insights into the importance of ASB screening and either sample type alone, is likely not sufficient to assess the microbiomes of both sites, and subsequently assess ASB/UTI risk; however, I have several concerns with it as written:

A principal concern with the results as illustrated is the lack of clarity as to what cutoffs were used for the genera included in Figures 1 and 2. There are over 50 genera in the EQUC figure and only 12 in the amplicon figure. Was a minimum relative abundance threshold applied for these figures? If so, please indicate. The second paragraph emphasizes that 25 genera were shared, perhaps including the 25 shared in these figures would add clarity. Minimally, please indicate what criteria were used for inclusion of genera in the figure legends.

Response: Thank you for pointing this out. The selection criteria were set to omit the lowest 5% abundant taxa, as described below (L 189-193).

"To avoid calculating the probability for genera that are at the limits of detection, where presence and absence of a taxa may be attributable to chance, we eliminated from this analysis (i) the lowest abundant genera, accounting for the lowest 0.5% of abundance in observed microbiomes, and (ii) genera that were detected in fewer than five samples.

**We also added text to the figure legends.
The 25 common genera are listed in Figure 3.**

Additionally, the figure quality (i.e. resolution) on all three primary figures is poor and makes some of the labels/names hard to read. I recommend updating them to higher resolution.

Response: Completed as requested. We have provided images with the best resolution that we can achieve.

Lastly, I noticed several inconsistencies and spots of poor clarity in the results section and recommend the authors revisit this portion after considering the minor comments/edits indicated below.

Response: We have attempted to resolve issues of clarity. See also our responses to the minor comments/edits below.

Minor comments/edits:

Vagina and urine are intermittently capitalized throughout the manuscript, please correct

for consistency.

Response: Done.

Abstract

No minor comments.

Introduction

No minor comments.

Methods

Line 110: Correct spelling of "Institutional"

Line 117: Change "delivery" to "deliver"

Line 124: Delete "a sterile"

Line 132: Change semicolon to an apostrophe

Line 174: Correct spelling of "perfect"

Be consistent with the capitalization of "Correlation"

Response: All done. Thanks for catching these errors.

Line 183: Should each P be accompanied by a subscript "t"?

Response: The reviewer is correct, and the equation has been updated.

Line 184: Should "condition" be "Sample type"?

Response: While 'sample type' in this manuscript does refer to 'vaginal' or 'urinary,' the mention of condition on L187 refers to conditional probability. While there is a reasonable argument for use of either 'condition' or 'sample type' here, we feel that its use in describing the conditional probability in this instance favors the more general 'condition' and that its use is clear in context.

Results

Line 192: Spell out OB.

Response: Done.

"Moderate" is indicated as a correlation range between 0.40-0.59 in the methods section, however in several instances, the correlations are equal to or greater than 0.60 and referred to as "Moderate". Please correct these instances.

Response: Done

Fix capitalization and quotations for consistency of "Moderate" also throughout the Results section.

Response: Done

Paragraph starting at Line 250: Please double-check the numbers, you state 8 of 64 genera then 16 genera in the subsequent sentence. The table the data is referencing indicate it should be 16 genera for this section.

Response: See next response.

Paragraphs at Line 243 and Line 258 appear to be copied with some numbers being incorrect. Please address and consider rewording the paragraphs in these sections so they are not so identical.

Response: We did discover some miss-entries due to the highly similar paragraph structure, which we have corrected. We have elected to retain the highly parallel structure to make results for different predictions as easy as possible for readers to scan and compare results.

Line 283: Add "ASB-associated" before "genera".

Response: Done

Paragraph at Line 288: The first two sentences are in contradiction. You state there was never a good chance and then in the subsequent sentence two genera are identified that had a good chance to be present.

Response: The reviewer is correct. The text has been changed to "...there was often not a good chance...."

Figures 1 and 2: Urine and Vaginal labels are cut off at the top of both figures and the Relative abundance label in Figure 2.

Response: The figures have been updated.

Tables 2 and 3: Add clarification of what the $p(\text{vaginal}|\text{urine})$ and $p(\text{Urine}|\text{vaginal})$, i.e. probably of taxa detected in urine to also be present in the vaginal microbiome, in a note or

indicate as each column label.

Response: This definition was provided in the section starting with L181. In the interests of clarity, we added footnotes to the tables.

Discussion

In the results portion of the discussion, I suggest reiterating the finding that the more relatively abundant taxa were more likely to have high probability of to occur in the other sample type.

Response: Good idea. The statement now reads (L 312-3):

“Vaginal and urinary microbiomes are, for the same patient, significantly correlated by their compositions, due primarily to the more relatively abundant genera. However, vaginal and urinary microbiomes are by no means equivalent.”

Reviewer #2 (Comments for the Author):

The manuscript authored by Wolfe et al. aims to compare the vaginal and urinary microbiomes of women during the first trimester of pregnancy, employing two distinct methodologies: EQUC and 16S rRNA gene (amplicon) sequencing. While I find the authors' perspective on the ASB concept and its pertinence in light of contemporary insights into the urinary microbiome and agree with it entirely, the study fails in its ability to fully address this facet of the objectives. This deficiency arises from: 1) the study design supported on the genus analysis; 2) Microbiome characterization methodology; 3) omission of a follow-up on the participants concerning ASB diagnosis and potential therapeutic interventions; 4) data analysis performed.

Following several flaws are detailed.

The study design supported on the genus analysis is an important flaw, namely when it's known that for the most prevalent and abundant genus/family different species (e.g. Lactobacillaceae 14 species of 4 genera; Corynebacterium 25 species; Gardnerella 9 species/genomospecies; Staphylococcus 14 species; Streptococcus 10 species) have been described and with different pathogenic potential.

Response: We are fully aware of this situation and we actually have the species level identifications for all the bacteria detected by EQUC and some of those detected by sequencing. However, we are simply asking whether the 2 different approaches can determine whether the 2 niches are similar enough to be used as proxies. If they do

not agree at the genus level, then they clearly won't agree at the species level - in fact, it would be worse. Thus, we respectfully disagree that this is a flaw in our analysis.

Furthermore, concerning the matter of methodology, the utilization of solely the V4 region of the 16S gene significantly undermines the capacity for precise bacterial identification, as it restricts the outcomes to genus-level categorization.

Response: It is not entirely true that the V4 region cannot provide species level identification. If one uses a Bayesian-based approach (such as BLCA [PMID: 28486927]), then species identification with some confidence can be obtained for some bacteria. However, as stated above, genus level identification is sufficient for this type of analysis. Thus, we respectfully disagree that this is a flaw in our analysis.

The methodology used for the EQUC also have limitations in capturing the full bacterial diversity and bacterial groups abundance. The plates used have low colony discriminatory ability for Gram positive and Gram negative bacterial genera. Moreover, the authors only characterized a distinct colony morphotype per plate.

Response: We do not know the basis of the reviewer's opinion, but we obtain excellent discriminatory ability. In fact, we can identify >70% of the bacteria detected at the genus level by 16S rRNA gene sequencing and almost that percentage at the species level by shotgun sequencing. Again, we respectfully disagree with the reviewer on this point.

Additionally, the employment of MALDI-TOF MS identification also reveals limitations, primarily due to prevailing databases being predominantly populated by clinically significant pathogens, thereby neglecting a substantial portion of the microbial species within the microbiome. In tandem with the inherent disparities in sample processing between these two methods, it becomes foreseeable that any meaningful correlation between their findings might prove elusive.

Response: Again, we respectfully disagree. While there are some disparities due to insufficiencies in databases, MALDI-TOF does a reasonably good job at the genus level and not bad at the species level. We just finished a study where we sequenced the genomes of 1000 isolates. For >90%, the MALDI identification and ANI analysis agreed.

In addition, regarding methodology, an important concern arises from the criteria used to calculate the likelihood of a particular genus being present in one sample and concurrently identified in the other. To achieve this, the authors opted to exclude genera with very low abundances from the analysis, specifically those accounting for the lowest 0.5% of the total abundance. Regrettably, this strategy led to the inclusion of a mere 11 out of 35 (31%)

genera from the EQUIC-detected microbiomes and 64 out of 428 (15%) genera from the amplicon-detected microbiomes. This approach, in effect, injects a noticeable bias into the analysis by discarding a significant portion of the bacterial community that was originally detected. Given the intrinsic nature of the urinary urobiome, where diversity is pronounced and many genera are naturally present in minimal abundances, the choice to analyze only 31% of the genera obtained through EQUIC and a mere 15% of the community identified through amplicon sequencing stands as an erroneous probability of the presence of a given genus in one sample to be identified in the other. It is my opinion that this analysis should be repeated with the majority of the identified community.

Response: As we responded to Reviewer #1, the selection criteria were set to omit the lowest 5% abundant taxa, as described below (L 189-193).

"To avoid calculating the probability for genera that are at the limits of detection, where presence and absence of a taxa may be attributable to chance, we eliminated from this analysis (i) the lowest abundant genera, accounting for the lowest 0.5% of abundance in observed microbiomes, and (ii) genera that were detected in fewer than five samples.

As such, we focused on the taxa that constitute 95% of the total abundance and observed that higher abundance does not correlate with prediction. If we had included the rarest taxa, the correlation would have been worse and thus strengthened our conclusion.

Throughout the manuscript, there are instances of conflicting ideas regarding the correlation between the two microbiomes. These disparities should be highlighted to clarify the study's perspective and its alignment with prior research findings. For instance, in the abstract, a sentence states, "Bayes Theorem quantified how well the vaginal microbiome could be used as a proxy for a patient's urinary microbiome and vice versa." However, in the concluding paragraph, it is stated, "While vaginal and urinary microbial compositions were significantly correlated for the same individual, they were by no means equivalent. The first trimester urinary and vaginal microbiomes are distinct enough to preclude their use as proxies of each other."

Response: We're sorry but we don't see how these two statements contradict each other. The first statement says that the Bayes Theorem was used to determine whether the vaginal microbiome could be used as a proxy for the urinary microbiome and vice versa. The second statement says that they could not be used as proxies for each other. One is about methodology and the other is about conclusion. However, to increase clarity, we have changed the first statement to now read (L 56):

"We used the Bayes Theorem to quantify how well the vaginal microbiome could be used as a proxy for a patient's urinary microbiome and vice versa."

Another conflicting notion is presented in the abstract: EQUC and amplicon sequencing are portrayed as complementary, unveiling analogous perspectives of the urinary and vaginal microbiomes in first-trimester pregnant women. However, if these methodologies indeed yield similar viewpoints, their complementarity might be questioned. It could be argued that true complementarity arises when their findings necessitate consolidation. Alternatively, if one methodology proves more elucidating, a discerning selection could be made to favor one over the other.

Response: We respectfully disagree with the reviewer's definition of "complementary." Here is the definition from the Oxford Languages: "combining in such a way as to enhance or emphasize the qualities of each other or another." In our experience, EQUC and 16S sequencing provide very similar but distinct views of the urobiome. They enhance each other. However, we have removed the word "complementary." The text now reads (L 58):

"Our findings provide evidence that EQUC and amplicon sequencing reveal similar views of urinary and vaginal microbiomes of first trimester pregnant women."

Specific points

Material and Methods

Line 137- details about media suppliers are missing. This is important to check the medium composition, e.g. the difference between Aerobic Blood agar and blood agar.

Response: EQUC details have been published many times before. For example, see reference #17.

Line 144- Details about the brand and model of Maldi-TOF MS equipment should be provided, as well as interpretation threshold used for considering the bacterial identification provided by the equipment.

Response: Ditto. To be clear, we have added reference #17 to the end of the sentence concerning MALDI.

Line 157-159 - which criteria was followed for considering 2 samples as discrepant?

Response: We have added the following criterion (L 161-2):

“Discrepancy was defined using the Bray Curtis Dissimilarity Index, where a value of 0.5 was considered sufficiently different to merit a third sample.”

The genus *Lactobacillus* is currently split into different genera. The authors should have used methodologies allowing *Lactobacillaceae* genus identification. The role of those genera and *Lactobacillaceae* species are being unveiled, suggesting different impacts in the microbiome of the urogenital tract.

"*Lactobacillaceae*" should replace "*Lactobacillus*" when mentioning several former *Lactobacillus* genera.

Response: We are well aware of the recent emendation of the genus *Lactobacillus*. The *Lactobacillaceae* species that are most prevalent and most abundant in the urinary tract remain in the genus *Lactobacillus*. These are *L. iners*, *L. crispatus*, *L. jensenii*, *L. mulieris*, *L. gasseri* and *L. paragasseri*. Thus, we respectfully contend that use of the term *Lactobacillus* remains most appropriate.

Results

Information on the number of colonies identified per plate should have been provided. More detailed information per sample should have been provided (e.g. total bacterial cells count; counts for different genera; reads per genus).

Response: Of course, we possess all of this information, and it could be provided. However, the table would be ponderous, and no reviewer has ever asked us for raw EQUIC data. It would be equivalent of putting all the raw sequencing reads in a supplemental data. We would be happy to deposit our raw culture data in a publicly available database, but no such database exists despite efforts by the urobiome community to obtain funding to establish such a database. Again, we respectfully decline the request.

Re: Spectrum02638-23R1 (First trimester 'clean catch' urine and vaginal swabs sample distinct microbiological niches)

Dear Dr. Alan J Wolfe:

Your manuscript has been accepted, and I am forwarding it to the ASM production staff for publication. Your paper will first be checked to make sure all elements meet the technical requirements. ASM staff will contact you if anything needs to be revised before copyediting and production can begin. Otherwise, you will be notified when your proofs are ready to be viewed.

Data Availability: ASM policy requires that data be available to the public upon online posting of the article, so please verify all links to sequence records, if present, and make sure that each number retrieves the full record of the data. If a new accession number is not linked or a link is broken, provide production staff with the correct URL for the record. If the accession numbers for new data are not publicly accessible before the expected online posting of the article, publication may be delayed; please contact ASM production staff immediately with the expected release date. You have communicated to me that the raw sequencing data for this manuscript are available as BioprojectID PRJNA1034602. Please include an appropriate Data Availability statement in the proofs.

Sincerely,
Kevin R. Theis
Editor
Microbiology Spectrum

Reviewer #1 (Comments for the Author):

The authors have appropriately addressed my prior concerns. I do recommend these minor corrections be made:

Results
Line 285: Change "ASB genera" to read either "ASB-associated" or "ASB-relevant".

Tables 2 and 3: Change "Genera" to lowercase.